# Current Pharmacological Treatment of Painful Diabetic Neuropathy: A Narrative Review

**DOI:** 10.3390/medicina56010025

**Published:** 2020-01-09

**Authors:** Valeriu Ardeleanu, Alexandra Toma, Kalliopi Pafili, Nikolaos Papanas, Ion Motofei, Camelia Cristina Diaconu, Manfredi Rizzo, Anca Pantea Stoian

**Affiliations:** 1Department of Surgery, University “Dunarea de Jos”, 800008 Galati, Romania; valeriu.ardeleanu@gmail.com; 2Department of Surgery, University “Ovidius’’, 900470 Constanta, Romania; 3Arestetic Clinic, 800098 Galati, Romania; 4Department of Surgery, Emergency County Clinical Hospital “Sf. Apostol Andrei”, 800578 Galati, Romania; 5Second Department of Internal Medicine, Diabetes Centre-Diabetic Foot Clinic, Democritus University of Thrace, University Hospital of Alexandroupolis, 681 00 Alexandroupolis, Greece; kpafili@hotmail.com (K.P.); papanasnikos@yahoo.gr (N.P.); 6Department of Surgery, “Carol Davila” University of Medicine and Pharmacy, 050474 Bucharest, Romania; igmotofei@gmail.com; 7Internal Medicine Department, Clinical Emergency Hospital of Bucharest, “Carol Davila” University of Medicine and Pharmacy, 050474 Bucharest, Romania; drcameliadiaconu@gmail.com; 8Biomedical Department of Internal Medicine and Medical Specialties School of Medicine, University of Palermo, 90133 Palermo, Italy; manfredi.rizzo@unipa.it; 9Division of Endocrinology, Diabetes and Metabolism, University of South Carolina School of Medicine Columbia, Columbia, SC 29209, USA; 10Diabetes, Nutrition and Metabolic Diseases Department, “Carol Davila” University of Medicine and Pharmacy, 050474 Bucharest, Romania

**Keywords:** diabetes mellitus, neuropathy, pain, pharmacological treatment

## Abstract

*Background and Objectives*: Distal symmetrical polyneuropathy (DSPN) is one of the most common chronic complications of diabetes mellitus. Although it is usually characterized by progressive sensory loss, some patients may develop chronic pain. Assessment of DSPN is not difficult, but the biggest challenge is making the correct diagnosis and choosing the right treatment. The treatment of DSPN has three primary objectives: glycemic control, pathogenic mechanisms, and pain management. The aim of this brief narrative review is to summarize the current pharmacological treatment of painful DSPN. It also summarizes knowledge on pathogenesis-oriented therapy, which is generally overlooked in many publications and guidelines. *Materials and Methods*: The present review reports the relevant information available on DSPN treatment. The search was performed on PubMed, Cochrane, Semantic Scholar, Medline, Scopus, and Cochrane Library databases, including among others the terms “distal symmetrical polyneuropathy”, “neuropathic pain treatment”, “diabetic neuropathy”, “diabetes complications”, ”glycaemic control”, “antidepressants”, “opioids”, and “anticonvulsants”. *Results*: First-line drugs include antidepressants (selective serotonin reuptake inhibitors and tricyclic antidepressants) and pregabalin. Second- and third-line drugs include opioids and topical analgesics. While potentially effective in the treatment of neuropathic pain, opioids are not considered to be the first choice because of adverse reactions and addiction concerns. *Conclusions*: DSPN is a common complication in patients with diabetes, and severely affects the quality of life of these patients. Although multiple therapies are available, the guidelines and recommendations regarding the treatment of diabetic neuropathy have failed to offer a unitary consensus, which often hinders the therapeutic options in clinical practice.

## 1. Introduction

Diabetic neuropathy is one of the most common chronic complications of diabetes mellitus [1]. It is defined as the presence of signs and/or symptoms of nerve dysfunction in patients with diabetes mellitus after exclusion of other causes [1,2]. The most frequent clinical manifestation is distal symmetrical polyneuropathy (DSPN), with a prevalence of 20–30% [2]. The precise pathophysiology of DSPN is multifactorial and complicated. Its main risk factors include diabetes duration, patient age, and vascular risk factors [3].

DSPN has an insidious course, characterized by chronic sensory loss with stocking and glove distribution [1]. However, it may also lead to chronic neuropathic pain [1]. DSPN treatment is complex, including both an optimal glycemic status, as well as the treatment of pain. Pain management in DSPN does not yet include specific medication to prevent or limit the reversibility of DSPN. Most often, clinical guidelines recommend symptomatic therapy, with the primary goal of pain reduction.

The present narrative review summarizes the current pharmacological treatment of painful DSPN. It includes brief references to emerging concepts and concerns such as opioid dependency and pathogenesis-oriented therapy (mainly α-lipoic acid), which are generally overlooked in many publications and guidelines.

## 2. Materials and Methods

We performed a review of the literature starting from 1990 by searching PubMed, Cochrane, Semantic Scholar, Medline, Scopus, and Cochrane Library databases for all observational studies, randomized clinical trials, and meta-analyses including the terms “distal symmetrical polyneuropathy”, “neuropathic pain treatment”, “diabetic neuropathy”, “diabetes complications”, ”glycaemic control”, “antidepressants”, “opioids”, and “anticonvulsants”, as well as their combinations regarding DSPN. All currently available original studies, abstracts, and review articles including systematic reviews and meta-analyses were examined. Case reports and letters to the editor were excluded. Publications in English were studied in full, whereas those in other languages only in abstract form.

## 3. Results and Discussion

### 3.1. Pharmacotherapy of Diabetic Neuropathy

There are several guidelines on the optimal pharmacological treatment of painful DSPN [4,5,6]. There is currently a general agreement on first-line drugs and other options.

### 3.2. Glycemic Control

An optimal early glycemic control may delay or even prevent DSPN in type 1 diabetes mellitus and prediabetes [7,8,9,10,11]. In type 2 diabetes mellitus, this is less effective [12,13,14,15]. Interestingly, specific glucose-lowering strategies may have different effects. In a post-hoc analysis of the BARI 2D clinical trial (Bypass Angioplasty Revascularization Investigation in Type 2 Diabetes), subjects treated with insulin sensitizers exhibited a reduced incidence of DSPN at 4 years, as compared with those receiving insulin or sulphonylureas [16].

### 3.3. Antidepressants

Duloxetine is a selective norepinephrine and serotonin reuptake inhibitor. It is used with a dose of 60–120 mg daily [5,17,18,19]. Duloxetine appears to improve the quality of life of patients with painful DSPN [20]. Its main metabolic adverse events include modest increases of fasting plasma glucose in both short- (12 weeks) and long-term (52 weeks) treatment, a modest increase in glycated hemoglobin A1c, and small non-significant weight gain in extension studies (52 weeks) [21]. Other common side effects include dry mouth, decreased appetite, sleepiness, sweating, and gastrointestinal problems [22]. Duloxetine was approved by the US Food and Drug Administration (FDA) for painful DSPN [23].

Tricyclic antidepressants may be used as well, especially amitriptyline. Amitriptyline appears to relieve pain in comparison to placebo in patients with DSPN [24] and seems non-inferior to pregabalin [25], gabapentin [26], and duloxetine [27]. Its precise mechanism of action remains to be clarified. Nonetheless, two hypotheses have been proposed: (1) inhibition of serotonin and norepinephrine reuptake [28] and (2) antagonism of *N*-methyl-d-aspartate receptors, which mediate hyperalgesia and allodynia [29,30]. Amitriptyline may be considered for DSPN at a single dose of 25 to 100 mg [31], although a dosage range of 25 to 150 mg per day has been suggested [28]. Amitriptyline has FDA approval only for the treatment of depression [32,33]. The side effects include dry mouth, water retention, constipation, and vertigo. Furthermore, clinicians should assess the QTc interval, to avoid the risk of torsades de pointes, especially among subjects with additional cardiovascular risk factors [33].

Although not formally approved, venlafaxine has been used as well [5], with evidence of effectiveness for the short-term management of painful DSPN [17]. It is a potent serotonin and norepinephrine reuptake inhibitor [32]. The proposed off-label dosage ranges between 75 and 225 mg per day in patients with DSPN [31]. Common side effects include sleepiness, dizziness, and mild gastrointestinal problems [34]. Patients should be regularly assessed for QTc prolongation [35].

### 3.4. Anticonvulsants

The leading agent of this class is pregabalin, a calcium channel subunit α2-δ binder. It has been approved by the US Food and Drug Administration (FDA) for painful DSPN [36]. The dosage ranges from 150 to 300 mg daily [36]. Many studies have reported favorable outcomes with more than 30–50% pain improvement [5,17,18,37,38,39]. However, not all pregabalin-related studies had positive results [5,17,19,39,40]. Untoward effects include water retention, visual disturbances, drowsiness, ataxia, euphoria, and vertigo [41].

Gabapentin also belongs to this class of agents [42]. It has the same therapeutic target as pregabalin [39]. Gabapentin has been reported to relieve pain among DSPN subjects [26,42,43,44,45], and combination with venlafaxine seems to provide additional benefit [46]. Although not formally indicated [47], guidelines of the American Academy of Neurology suggest the use of gabapentin for the treatment of DSPN as a second-line therapy after pregabalin [31]. The recommended off-label daily dosage for DSPN subjects ranges between 900 and 3600 mg [31]. In patients with chronic kidney disease, the doses of gabapentin and pregabalin need to be adjusted: the renal clearance (CrCl ≤ 30 mL/min) requires dose adjustment, with a recommended dose of gabapentin of 300 mg daily and pregabalin of 75 mg daily [48]. Gabapentin has been associated with sleepiness, dizziness, suicidal behaviour, withdrawal-precipitated seizure frequency, multi-organ hypersensitivity, systemic symptoms, and drug reaction with eosinophilia [47].

Other anticonvulsants have been used as well (carbamazepine, lamotrigine, lacosamide, etc.), with variable results [5,17,48,49,50].

### 3.5. Opioids

Tapentadol is a centrally acting opioid analgesic that exerts its analgesic effects by inhibiting the μ-opioid receptor and noradrenaline uptake [51]. The FDA approved prolonged-release tapentadol for painful DSPN, based on data from two clinical trials in which patients titrated with an optimal dose of tapentadol were arbitrarily requested to continue with that dose or to change it with placebo [52,53]. However, both studies used an enriched design for patients who responded to tapentadol, and therefore the results obtained are not generalizable. Importantly, a recent systematic review and meta-analysis by the Special Interest Group on Neuropathic Pain within the International Pain Study Association found that evidence supporting the effectiveness of tapentadol in reducing neuropathic pain is inconclusive [5]. Finally, addiction is a serious concern with long-term opioid use [32].

### 3.6. Topical Treatment

Capsaicin is a natural alkaloid [54]. This is thought to desensitize afferent Aδ and C fibers [55]. Two forms of capsaicin are available for the treatment of painful DSPN: a low-dose cream (0.075%) [31] and a high-dose (8%) patch (Qutenza, Acorda Therapeutics, Ardsley, NY, USA) [56]. A series of studies have provided evidence of pain relief among subjects who received capsaicin cream in comparison to the vehicle [57,58,59,60], although results were inconsistent [61,62,63]. An older study even provided evidence of equal efficacy of capsaicin cream (0.075%) to oral amitriptyline for painful DSPN [64]. In agreement with the guidelines of the American Academy of Neurology [31], a dosage of 0.075% four times daily is recommended for the treatment of painful DSPN. A notable disadvantage is local adverse effects— mainly stinging, burning, and erythema [65].

Patches with capsaicin 8% have also been studied in painful DSPN, with conflicting results [66]. Nonetheless, some evidence points to superiority in comparison with oral pregabalin, duloxetine, and gabapentin for pain relief among DSPN patients [67]. Although not formally approved [68], a single application of this patch may provide up to 12 weeks of pain relief [69]. This should be performed under specialist supervision, with appropriate local anesthesia and monitoring for blood pressure increase, especially during the first hour following application [69].

The 5% lidocaine plaster is licensed for postherpetic neuralgia in approximately 50 countries around the world and for localized neuropathic pain in 11 Latin American countries [70]. The lidocaine molecule is a voltage-gated sodium channel inhibitor which blocks abnormally functioning neuronal sodium channels (in the dermal A-δ and C fibers) [71]. Studies have provided evidence of pain relief in DSPN patients comparable to pregabalin [72,73], amitriptyline, capsaicin, and gabapentin [73]. A maximum of three lidocaine patches 5% can be applied to intact skin once for 12 h within a 24-h period [70]. Adverse events include mild and transient application-site reactions—mainly erythema, edema, and a burning sensation [74].

Finally, topical clonidine, a presynaptic α-2 adrenergic receptor agonist with antinociceptive activity, was associated with pain relief in DSPN in a small number of studies of low-to-moderate quality [75]. Clonidine gel 0.1% may be administered in single doses of 0.65 g of gel (0.65 mg of clonidine), three times daily so that the total daily dose should not exceed 3.9 mg for both feet. The administration is associated with only mild skin-site reactions [76].

### 3.7. Pathogenesis-Oriented Treatment

α-Lipoic acid is a natural thiol with potent antioxidant properties, and is used as a dietary supplement. In studies evaluating this indication, it has been administered orally at doses between 600 and 1800 mg and intravenously at 600 mg per day for 3 weeks, excluding weekends [77,78,79,80,81,82]. Both formulas have been recently characterized by the FDA as safe and effective treatment options for painful DSPN [83]. The efficacy of the daily administration of a 600 mg intravenous formula for 3 weeks was investigated in a meta-analysis of four randomized, placebo-controlled trials (ALADIN I, ALADIN III, SYDNEY, NATHAN II) comprising 1258 participants with painful DSPN [82]. It was shown that α-lipoic acid improved positive neuropathic symptoms (24.1% in favor of α-lipoic acid versus placebo), but failed to alter the neuropathy impairment score significantly [82,83].

Finally, actovegin is a deproteinated, pyrogen- and antigen-free ultrafiltrate obtained from calf blood. It appears to ameliorate oxidative stress and to exhibit antiapoptotic properties [79]. In a study conducted by Ziegler et al., 567 patients were treated with actovegin (n = 281) versus placebo (n = 286). After 6 months, compared with placebo, actovegin was associated with better improvement of neuropathic symptoms, but not pain specifically (OR [95% CI] of 1.73 [1.21–2.48] for patients treated with actovegin and 1.94 [1.33–2.84] for patients placebo treated). Actovegin treatment (20 daily intravenous infusions of 2000 mg/day followed by three actovegin tablets of 600 mg each for 140 days) was associated with a clinically significant response in neuropathic symptoms and vibration perception threshold in patients with symptomatic DSPN, with a good safety profile [84]. Further studies are needed to demonstrate the efficacy of α-lipoic acid and actovegin in painful DSPN. Current pharmacological agents for painful DSPN are summarized in Table 1.

## 4. Conclusions

DSPN is a burden both from diagnostic and treatment perspectives. It is still under-diagnosed and undertreated.

As summarized in this narrative review, first-line treatment options include pregabalin and duloxetine, with gabapentin seeming a reasonable alternative to pregabalin treatment. Second- and third-line drugs include opioids and topical analgesics. Indeed, opioids are an effective alternative in the treatment of neuropathic pain; however adverse reactions and addiction concerns limit their widespread use. More recently, research has extended beyond symptom alleviation. Pathogenesis-oriented treatments, including α-lipoic acid and actovegin, appear promising, but results need to be confirmed in more extensive trials.

## Figures and Tables

**Table 1 medicina-56-00025-t001:** Current pharmacological agents for painful distal symmetrical polyneuropathy (DSPN) [5,6].

Pharmacotherapy	FDA Approval for DSPN	Daily Dosage	Untoward Effects	Comments
**Antidepressants**				
Duloxetine	Yes [23]	60–120 mg/d [5,17,18,19]	Xerostomia, decreased appetite, somnolence, sweating, gastrointestinal discomfort [22]	Appears to improve the quality of life of patients with painful DSPN [20]
Amitriptyline	No [33]	25–100 mg/d [31]	Xerostomia, water retention, increased appetite, weight gain, constipation, vertigo [33]	Appears non-inferior to pregabalin [25], gabapentin, [26] and duloxetin [27] in painful DSPN; monitor QTc interval [33]
Venlafaxine	No [31]	75–225 mg/d [31]	Somnolence, dizziness, mild gastrointestinal problems [34]	Evidence of effectiveness for short-term management of painful DSPN [17]; monitor QTc interval [35]
**Anticonvulsants**				
Pregabalin	Yes [36]	150–300 mg/d [36]	Water retention, visual disturbances, somnolence, ataxia, euphoria, vertigo [41]	DSPN-related pain improvement of >30%–50% has been reported [5,17,18,37,38,39]
Gabapentin	No [47]	900–3600 mg/d [31]	Somnolence, dizziness, suicidal behaviour, withdrawal-precipitated seizure frequency, multi-organ hypersensitivity [47]	DSPN-related pain relief has been repeatedly reported [26,42,43,44,45,46]
**Opioids**				
Tapentadol extended release	Yes [52,53]	100–500 mg/d [51]	Dizziness, somnolence, headache, fatigue, gastrointestinal problems [51]	Inconclusive pain reduction [5]; addiction concern [32]
**Topical treatment**				
Capsaicin cream	No [32]	0.075% four times/d [31]	Skin-site reactions [63]	Equal efficacy to amitriptyline [62]
Capsaicin 8% patch	No [68]	One application every 3 months [66]	Skin-site reactions [66]	Conflicting results in DSPN [66]; application can be painful [66]; monitor for transient blood pressure increase for at least one hour following application [66]
5% lidocaine plaster	No [70]	Maximum 3 lidocaine plasters 5% can be applied to intact skin once daily for a period of 12 h [70]	Skin-site reactions [74]	Comparable efficacy to pregabalin, amitriptyline, capsaicin, and gabapentin [72,73]
Clonidine gel 0.1%	No [75]	Single doses of 0.65 g of gel, three times daily [76]	Skin-site reactions [76]	Pain relief in a small number of studies of low-to-moderate quality [75]
**Pathogenesis-oriented treatment**				
α-Lipoic acid	No [77]	600–1800 mg orally or 600 mg/d intravenously for 3 weeks, excluding weekends [77,78,79,80,81]	Nausea, vomiting, abdominal discomfort, diarrhea [77]	FDA statement: safe and effective treatment option for painful DSPN [82]

Legend: FDA: Food and Drug Administration.

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
