# Peer review of "Current Pharmacological Treatment of Painful Diabetic Neuropathy: A Narrative Review"

_medicina, 2020, doi:10.3390/medicina56010025_

Round 1

Reviewer 1 Report

I have seen this manuscript before and there were no major changes. If you insist on publishing this manuscript, I suggest you rename it a narrative review because it seems that there is not enough reporting on how systematically you reviewed the literature.  There is also no attempt to assess the quality of articles selected and you have not even mentioned how many were selected and what are the characteristics of the studies.

Author Response

Dear Reviewer, 

We thank you for the comments. As rightly suggested by you, this is, indeed, a narrative review, as already stated in the title. We have now added the word "narrative" in the Abstract, in the Introduction and in the Conclusion part, in order to emphasise the nature of the review.   Obviously, detailed descriptions of study characteristics are beyond the scope of narrative reviews. We appreciate your comments, we have now added more information in the Material and Methods section, to increase the clarity.  All changes are made in yellow.

Reviewer 2 Report

Thank you for addressing our suggestions. All the best. 

Author Response

Thank you very much!

All the suggestions were added.

This manuscript is a resubmission of an earlier submission. The following is a list of the peer review reports and author responses from that submission.

Round 1

Reviewer 1 Report

This manuscript was presented as brief review but this is in essence a poorly conducted systematic review which lacks reporting of how articles were selected and is not adhered either to PRISMA guidelines or Cochrane Collaboration methodology. It is not clear how the main table of results was produced. What evidence is there for the effectiveness of the drugs reviewed?

There was also no attempt to assess the quality of the articles selected.

I suggest that the authors refocus the manuscript on the FDA approved drugs or even the main drug used in reducing painful DN and evaluate the evidence of the effectiveness of these drugs in a systematic approach. 

Author Response

Dear Reviewer 1, 

Thank you for your relevant comments. 

In our article, we wanted to review the medication involved in the treatment of DSPN pain. We did not want a very precise and focused approach especially of any of the molecules because we want to address and aiming to help general practitioners. We wanted to be practical in briefly reviewing current therapeutic choices rather than focusing on one or two drugs and writing a systematic review on them. Regarding the main table, we wanted a broader coverage than just FDA-approved drugs.

For the changes I have made, please see the attachment.

Thank you so much for your suggestions.

Reviewer 2 Report

Sentence #54, (DSPN is usually painless.) In early stages this is inaccurate. It also conflicts with sentences following. I would remove this statement.

Sentence #78, (Optimal early glycemic control....) I would add to this in prediabetes also. 

Sentence #110, Extra W before word It at end of sentence.

Sentence #119, Capitalize Neurology

Sentences #122-124, When considering side effects of gabapentin you have to take into account the person with end-stage-renal-disease and the side effects if given prescribed dose.

Sentence 187. Delete "it" between DSPN and is

Author Response

Dear Reviewer 2,

Thank you for the relevant comments, they improve the article quality.

All the changes you have reported are already done, please see the attachment.

Thank you!  

Reviewer 3 Report

This brief review of the treatment of painful diabetic neuropathy can be useful as a quick reference for a large category of practitioners. Needs to be a little bit more focused on the pain itself.

Abstract:

Line 25-26: I would not say that the most difficult aspect of DSNP is making the diagnosis but identifying an effective treatment is

Line 26: correct spelling, “diagnosis” instead of “diagnose”

Line 29-30: opioids dependency is indeed a significant and growing concern but I think the statement “it sheds more light on emerging concepts or concerns” is exaggerated, not enough supported in the text of this short review

Line 38-38: say “while potentially effective” instead of “while effective”, the quality of evidence for opioids is low (may see Bialas P et al, Eur J Pain, Oct 2019)

Maybe a few words about pain description in the Introduction (how disabling can it be) and Material and methods (how is it assessed) would be useful.

Material and Methods

Line 65: the authors report that they did a systematic literature search starting from 2009 but there are multiple older references, including 1990s

Results and Discussion

Page 73-74: the guidelines in the reference mentioned (4) refer to neuropathic pain in general, not DSPN

Table 1 and text:

For Amitriptyline, would add weight gain due to increase appetite

Anticonvulsants: line 110, delete “W” to read”…It has been…”

Pathogenesis-oriented treatment:

Line 175-179: the authors need to refer to the relevance of the 4 year study of oral A-lipoic acid specifically for the painful DSPN, otherwise would remove these 2 sentences

Line 180-185: similarly, the authors need to refer to the relevance of the Actovegin study, specifically for the pain of DSPN, as study outcome refer to a composite score of 4 neuropathic features, not pain independently, so it is probably difficult to attribute the effect of actovegin on pain itself

Author Response

Dear Reviewer 3, 

Thank you for your accurate comments; they are improving the article quality.

The purpose of our article was to review the medication recommended in the treatment of painful DSPN. That is why we choose to discuss both the treatment of pain and other factors involved in its control (glycemic control for example), knowing that poor metabolic control and diabetes progression are factors that can influence and aggravate DSPN.

Regarding materials and methods, we revised the articles between 2009 and 2019 but some reference like DCCT trial (1995) or Quilici, Wernicke studies are the reference in the field. If you consider that a longer search period should be mentioned, please let us know. 

For the next comments, we made all the corrections as you recommended, please see the attachment.

Thank you very much for your comments.

Round 2

Reviewer 1 Report

I have not noticed that my previous comments were addressed. 

Author Response

Thank you very much.I have made all the corrections. Please find the second corrected version of our article.

Reviewer 3 Report

Line 53: Eliminate "it" from "DSPN it" 

Line 61-63: Would use "It includes brief references to ... opioid dependency and ... feasible pathogenesis-oriented therapy" instead of "It sheds more light on emerging concepts and concerns, such as opioid dependency. It also summarises knowledge on feasible pathogenesis-oriented therapy" 

Materials and methods: there are multiple articles published before 2009 cited in the manuscript. The authors should probably extend the duration of "systematic review of the literature" starting with 1990 (oldest reference provided). Otherwise, the explanation "Some biographical references and studies from 2009 (?) were considered as relevant studies in the field of the subject." makes the "systematic review" statement less credible

Results

Line 171-192: the authors still should make the "pathogenesis-oriented therapy" paragraphs more relevant to the painful diabetic neuropathy rather than discuss effect on neuropathic symptoms in general. It can be shortened 

Author Response

Thank you very much.I have made all the corrections.